# State *Ibuism* and Women's Empowerment in Indonesia: Governmentality and Political *Subjectification* of Chinese *Benteng* Women

Vinny Flaviana Hyunanda [1], José Palacios Ramírez [2], Gabriel López-Martínez [3,*] and Víctor Meseguer-Sánchez [4]

1   Department of Social Science, San Antonio Catholic University of Murcia, 30107 Murcia, Spain; vfhyunanda@ucam.alu.edu
2   Department of Psychology, San Antonio Catholic University of Murcia, 30107 Murcia, Spain; jpalacios@ucam.edu
3   Department of Contemporary Humanities, University of Alicante, 03698 Alicante, Spain
4   International Chair of Social Responsibility, Catholic University of Murcia, 30107 Murcia, Spain; jvmeseguer@ucam.edu
*   Correspondence: gabriel.lopez@ua.es; Tel.: +34-965-902-033

**Abstract:** This paper examines how the patriarchal understanding of "women's empowerment" in Indonesia instrumentalizes the notion of *Ibu*, a social construction of womanhood based on a societally determined idea of domestication and productivity. Through the establishment of a saving and lending cooperative, a group of Chinese *Benteng* women was subjected to a neoliberal development project that operated on the basis of a market-driven society and promoted a "gender mainstreaming" discourse to enhance this participatory project. They were introduced by a women's NGO as their broker. The notion of "women's empowerment" inspired a governmental operation aimed at these women, promoting the particular qualities of the dutiful housewife, devoted mother, and socially active member of Indonesian society. These characters were distinguished by their high level of devotion to community volunteering and to the state's apolitical project, thus depoliticizing and deradicalizing the feminist view of women's empowerment; this was simultaneously balanced with the promotion of the traditional gender roles of wife and mother. Such a discourse also molds women's desires to voluntarily subscribe to such a social construction of womanhood and, at the same time, circumvents objections to any form of women's subordination reproduced by the same rhetoric of "women's empowerment". By employing an ethnographic methodology, this article argues that the patriarchal view of "women's empowerment" emerged as a deceitful doctrine to prompt Chinese *Benteng* women into internalizing certain qualities according to the gendered conception of womanhood in Indonesia. This article concludes that the patronizing and dominating aspects of State *Ibuism* have normalized Indonesian society's expectations and desires with regard to women's empowerment.

**Keywords:** subjectification; governmentality; women's empowerment; cooperative; feminism

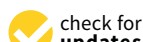



## 1. Introduction

The discussion of women's involvement in development has been a fundamental aspect of the policy-making process. In addition to the increasing use of "women's empowerment" as a bottom-up discourse, which initially emerged as an alternative to the top-down model, the term is often seen as a way to increase women's participation. For decades, "women's empowerment" has been exhaustively exploited by different agencies at various levels to win development contracts, especially for projects that aim to emancipate women [1]—and in Indonesia this is no different. The incorporation of women into Indonesia's development language is frequently associated with "gender mainstreaming policies" that intend to increase women's participation through "empowerment" and subsequently have them contribute to national growth. As a result, there is perceived to be

an urgent need to insert "women" into the strategic national agenda by institutionalizing "women's empowerment" as a gender mainstreaming strategy [2–5], conceiving "women" as a weak economic unit to be enhanced through "empowerment".

The importance of including women in the dominant development framework was disseminated through the Women in Development (WID) paradigm, coined by Ester Boserup in 1970 in her work Women's Role in Economic Development. As an economist, Boserup criticized a development process that fails to effect positive changes in women's lives. Furthermore, she argued that such failures are caused by women's marginalization from the economic system, since they are confined to their reproductive role and non-wage labor. Since women have been left out of the development process, they should be included in development efforts by being integrated into the existing mechanism. For decades, the WID paradigm has been a dominant framework that many development agencies and organizations work with. This approach underlines that women's contributions to development were undervalued due to their lack of participation. Therefore, women's participation is prescribed as a corrective strategy to bring women into development through the creation of "opportunities" that engage them and in theory will promote greater equality in terms of receiving development benefits. Consequently, the language of women's participation in development is interpreted as an economic language that understands women's empowerment not only as an instrument of reproductive work, but also as an untapped source of productive labor.

The emergence of the Gender and Development (GAD) paradigm (an approach that views gender relations as asymmetrical in that men and women are socially constructed as dominant and subordinate, respectively; this is reproduced over time and strongly embedded in the social, cultural, and historical context of a particular society—here, unequal power relations are the core issue to address and "women's empowerment" is conceptualized as a strategic vehicle to overcome such inequality) in the early 1980s, an alternative approach to the Women in Development paradigm, allowed some Indonesian feminists to start focusing their analysis on gender relations rather than simply gender roles. This approach aims to transform unequal gender relations by viewing women as agents of change rather than passive beneficiaries, in a consolidated effort to challenge the unequal distribution of power. Eddyono [4] argued that, by introducing the concept of "gender" through this approach in the Indonesian context, it is more feasible to discuss the power relations between men and women without the explicit "feminist" language that is generally viewed as a "Western" concept [6]. Particularly since Indonesia's political reform in 1998, which marked the fall of the New Order authoritarian regime, the phrase "*pemberdayaan perempuan*", as an official translation of "women's empowerment", became increasingly appealing, in addition to the term "gender" as a less intimidating concept; this was also marked by the establishment of the Ministry of Women's Empowerment, replacing its predecessor, the State Ministry of Women's Affairs. Despite such efforts, Indonesia's bureaucratically designed five-year national development plan (RPJMN)) deploys a homogeneous construction of "women" in state-sponsored empowerment programs, framed within the broader "gender mainstreaming" agenda. This process is stipulated in Law No. 17/2007, which requires that all national and local government bodies consider "gender" in their development planning, budgeting, and implementation and achieve results that foster "gender equality". Therefore, what was previously conceptualized as "*pemberdayaan perempuan*" was replaced by gender equity and equality, which makes "*pengarusutamaan jender*"—the official translation of "gender mainstreaming"—the overarching policy framework, explicitly mentioned in every national development plan as the key strategy to "reduce gaps between the Indonesian male and female population in accessing and obtaining benefits from development, as well as improving participation and controlling the development process" (RPJMN 2014, p. 10). Subsequently, this "gender mainstreaming" remains important in RPJMN 2015–2019 as a crucial mechanism to ensure the "improvement of life quality and women's role in development [ . . . ]" in order for Indonesia to be a "competitive nation". As a result, the "gender mainstreaming" strategy involves countless

activities intended to promote women's engagement in development activities that would empower them economically.

The term "women's empowerment" gradually lost its radical roots and became a simple buzzword for many development projects. The phrase "women's empowerment" and the term "gender mainstreaming strategy" are used as interchangeable terms to emphasize women's participation in the economic arena. Such a reductionist view has triggered strong criticism from some feminists who point out the conceptual departure towards an apolitical approach that deradicalizes the core of the gender inequality discourse [7–9]. These feminists argue that "women's empowerment" is understood more as a technical term that lacks key links with power analysis and fails to challenge power relations, and thus can easily become co-opted to serve an economic growth-based development model. Similarly, Biewener and Bacqué [10] criticized the fact that power has been taken out of the empowerment equation, making this alternative vision mainstream and depoliticizing the approach according to the principle of market-driven neoliberal policies. In a deradicalized way, power here implies that women's agency has been constructed as the ability to make a rational choice to profit from opportunities or resources to enhance one's wellbeing in a competitive environment [10,11]. Batliwala [7] pointed out that empowerment is now seen as a "magic bullet" for the alleviation of poverty rather than a way of interrogating the complex dynamics of a social process. Conversely, by converting political and social problems into market terms, "women's empowerment" becomes an attractive technical tool of a neoliberal development intervention, which deradicalizes the notion and focuses on individual participation and self-interested action.

This study aimed to unravel the conditions in which a patriarchal gender construction of womanhood prevails in post-reform Indonesia and to determine how it operates by deploying the rhetoric of "women's empowerment" as a way to govern. The findings were drawn from ethnographic fieldwork in Tangerang in 2018, during which data were gathered through participatory observation in various formal and informal events with different groups. The authors were extensively involved in Chinese *Benteng* women's everyday lives and paid close attention to their social context and how they interpreted and assigned meaning to it [12,13]. This fieldwork was part of a larger research project that aims to critically unpack the social inclusion approach towards this community, through operationalizing the technology of women's empowerment. Throughout the analysis process, the idea of women's empowerment appears to be used as a ploy to control Chinese *Benteng* women, with the creation of *Ibu* identity, in particular, held up as a desirable goal to suit the gendered conception of Indonesian womanhood. Furthermore, the controlling aspect of State *Ibuism* [14] as the official construction of Indonesian womanhood has normalized society's expectations for women's empowerment as a productive exercise.

In the following section, we contextualize the Chinese *Benteng* community as the subject of this study. We note that their experience of poverty is exoticized and explain how their involvement in a women's empowerment project started. Furthermore, we elaborate on Indonesia's formal construction of womanhood through a long history of state-engineered organization and explore how an idealized conception of womanhood is used to control people. We then analyze how the popular notion of "women's empowerment" is instrumentalized by market-driven development projects to cultivate particular qualities that would serve the neoliberal aid regime.

## 2. Theoretical and Empirical Background: Locating the Research Context

### 2.1. Chinese Benteng Women and the Exoticization of Poverty

For centuries, Chinese Indonesians have been constantly problematized by the ruling regimes and represented as "the Other" in Indonesia's nation-building trajectory. Various scholars [15–18] noted that, long before colonization, Chinese communities in the archipelago had been well integrated and culturally amalgamated with the local population through intermarriage [19,20]. This further led to the distinction between *Totok* Chinese and *Peranakan* Chinese in Indonesia. *Totok* literally means "pure", while the *Peranakan* group

refers to local-born or "mixed-blood" people. The *Peranakans* were the product of intermarriage between Chinese male immigrants and local women [17]. The Chinese *Benteng* people form one such community that historically shaped the culture and identity of Tangerang, a district on the outskirts of Jakarta [21,22]. Due to the widening economic gap between Jakarta and the surrounding regions, Chinese *Benteng* people are often stereotyped as lazy and illiterate and their lifestyle dismissed as backward and improper. Moreover, their distinctive physical features distinguish them from generic Chinese Indonesians, who are represented as "wealthy" and "fair-skinned". This contributes to derogative terms such as "poor Chinese" and "black Chinese" being used for *Benteng* people. Despite not every Chinese *Benteng* person conforming to such derogatory stereotypes, some live in a disadvantaged situation, particularly those who reside in poor *kampung* where public services are often unavailable.

In the 1970s, Jakarta's outskirts were called *kampung*, in contrast with Jakarta's representation as a *kota* (city); road infrastructure was extremely limited and this area could only reached by a horse-pulled cart. In the colonial era, the term *kampung* was associated with a native settlement, characterized as an underdeveloped area where the inhabitants were perceived as poor, uneducated, backward, and having poor hygiene [23]. Because of that, and linking their physical similarities with the native population, Chinese *Benteng* are often termed "Chinese *Kampung*", marking them out as different from the other Chinese in the city. In *Kampung* Wetan (a pseudonym; *Kampung* Wetan is one of many *kampungs* that comprise the administrative area of *Kelurahan* Kembangan in Tangerang Municipality, where the ethnographic field work was carried out), many Chinese *Benteng* people live on sparsely vegetated land, with houses that resemble those of natives but remain recognizable due to some Chinese attributes and symbols such as altars, incense, and joss paper. In the colonial era, *Kampung* Wetan was private land where Chinese *Benteng* people took up jobs in sugar cultivation. Following the end of colonialization, Chinese *Benteng* people's settlement pattern changed and, because of that, the concept of land ownership registration was generally foreign to them.

Chinese *Benteng* people are generally poor and so are often easy targets of illegal levies from street-level bureaucrats, who are complicit with the state administration. They are also the frequent object of media coverage that exoticizes their poverty. Their situation was worsened by industrialization in Tangerang during the 1990s, which contributed to massive land conversion as many sold their land "under the table" at a very low price to companies or mafias. Due to this, some of them moved to other areas, purchased other land, built new houses, started new settlements, or simply rented. Consequently, because of this conversion, many of them stopped farming and sought other alternatives to generate income. However, due to various factors such as a low level of education and a lack of physical mobility, they remain trapped in low-paid and low-skilled jobs. The men mostly work as parking attendants, pedicab cyclists, street hawkers, and factory laborers; the women are mostly housewives but may also work as laundresses or door-to-door ready-made meal or snack sellers.

Many Chinese *Benteng* families in *Kampung* Wetan follow Confucian teachings as the main pillar of their upbringing. However, this often centralizes regulating women's position and role in the household—rather than respecting their rights—which obliges a woman to be obedient to her father before marriage and dutiful to her husband after, and to be subservient to her son if her husband dies. In many Chinese *Benteng* families, women have multiple disadvantages in comparison with their male counterparts. Due to the belief that women should join their husband's family, wealthier Chinese *Benteng* families often prioritize men over women in terms of family assets. Consequently, most women often rely financially on their husbands due to a lack of family's wealth. Generally, Chinese *Benteng* girls and adolescents are taught to care for their parents more than their male siblings. This kind of upbringing makes them closer to her parents and causes them to spend more time in the house compared to boys, who are relatively free to spend time outside of the house.

In a contemporary situation, there are no stark differences between girls and boys in terms of schooling. Unlike 20 years ago, Chinese *Benteng* parents send both girls and boys to schools, even if they often face financial difficulties in paying extra school fees. Besides formal education, young and adolescent girls are expected to help their mothers with light household chores, including taking care of their younger siblings. While boys are often allowed to make a mess in the house and go play outside for the whole day, girls are required to be "tidy", as is appropriate and desirable for a "good" girl. Girls must also keep the house in order and not bring any trouble to their parents. Some poor families would prefer that girls find jobs or get married once they finish high school. This situation is apparent in *Kampung* Wetan, where conservative parents generally prefer to marry their daughters off as a way to ease the family's economic burden by shifting the responsibility for their daughter to another family. Up until ten years ago, more conservative families preferred to groom their daughters to be a "desirable wife" rather than encouraging them to pursue higher education, as this was considered costly and less desirable in a potential marriage partner. Most Chinese *Benteng* women above 45 years old stopped going to school before they completed their basic education because of poverty. For them, basic numeracy and reading are enough. Some other women between 25 and 40 years of age dropped out of high school and decided to get married.

The life of Chinese *Benteng* people is illustrated by various writings [24–26]. Purwanto [25] points out that Chinese *Benteng* people's multidimensional poverty is generally caused by low earnings and a lack of formal administration documents, which prevents them from accessing public services. Such a combination combines with the corrupt system run by a partnership between street-level bureaucrats and local brokers to maximize certain individuals' profit. In April 2010, *Kampung* Wetan made the national news because Tangerang city government planned to evict the local community from their settlement area, Cisadane riverbank, to develop a green zone and river normalization project. For months, the community complained that the government never informed them about the plan. They felt that they were never consulted by the local government, and a group of local residents went to the national parliament in Jakarta to voice their concerns, which, unfortunately, did not lead to favorable results. As the city government went ahead with the plan, deploying local police troops to destroy their "illegal" settlement, the community resisted, revolted, and rallied for a few days to safeguard their *kampung* from demolition. Their resistance was successful and the eviction plan was put on hold. However, they remain alert and are worried that a similar eviction might happen in the future.

### 2.2. The Koperasi Pelita as "Women's Empowerment" Platforms

After the eviction incident, the residents of *Kampung* Wetan, who are mostly Chinese *Benteng* people, became extremely suspicious of strangers, and their relationship with the government got even worse. This normalized the stereotype that Chinese *Benteng* people are a "socially isolated" community in *Kelurahan* Kembangan. Responding to that image of "social isolation", which is connected with their poverty, the Women's Resource Centre (WRC)—a Jakarta-based NGO—approached them to be beneficiaries of a donor-funded project on social inclusion by forming a cooperative ("*koperasi*") as their empowerment platform. After seeing online news and YouTube videos about the 2010 eviction incident, the WRC came to *Kampung* Wetan in early 2015 to carry out an assessment of social exclusion in the Chinese *Benteng* community and proposed acting as a donor. According to this NGO, which already has longstanding experience working with poor urban women, a *koperasi* would be a suitable solution that would give them beneficial activities to fill their spare time and overcome boredom by giving them social activities and a new network. The NGO theorized that, through the establishment of *koperasi* as inclusive women's empowerment platforms, Chinese *Benteng* women would become more socially active and involved in community activities.

Generally, in many women's groups in Indonesia, an *arisan* is the most popular way to get together (a form of microfinance in Indonesian culture). Forming an *arisan* group

is an attractive way to maintain social cohesiveness by pooling funds [27–29]. Another form is *kelompok simpan pinjam* (saving and loan groups), which have a more explicitly economic aspect and are considered to be a type of credit institution [30]. The combination of the economic philosophy of the *kelompok simpan pinjam* and the social aspect of the *arisan* led to the creation of a *koperasi simpan pinjam* (a savings and loan cooperative), which eventually became an effective organizing tool in all the women's projects run by the WRC. After weeks of approaching the Chinese *Benteng* women in *Kampung* Wetan, the WRC was finally successful at teaching them how to form their own *koperasi simpan pinjam*, which was named *Koperasi* Pelita (a pseudonym). This *koperasi* was also used to familiarize women with the concepts of mutual collaboration and self-help, which WRC found that these women were lacking.

Under the direction of WRC staff, who came to *Kampung* Wetan on a weekly basis, about 25 women gathered and elected a leader as well as a small coordination team to run the *koperasi*'s financial operations. The WRC also identified some inspiring and active women to be local leaders of other activities, such as lessons in *Cokek Sipatmo* (a traditional dance), baking classes, helping with registration documents, and facilitating *audiensi* (a meeting between a government representative or bureaucrat and citizens) with the local government. Within three years, the *Koperasi* Pelita successfully attracted widespread attention in *Kampung* Wetan, especially from government officials. Their savings and lending activities were successful at eliminating the presence of intimidating loan sharks. The *koperasi*'s economic units, "Pelita Cake" and "Pelita Dance Group", also started to generate a small profit for their members. Pelita Dance Group, in particular, has become a source of local pride and is acclaimed by officials due to its frequent performances as part of various events. Consequently, the WRC's presence in *Kampung* Wetan was praised, not only by the women but also by the officials, as their *kampung* become *harum*. Some active women also become more popular than others, as they were regularly invited by government officials to attend important meetings and were appointed as government cadres (representatives).

In the past few years, the WRC, along with some *Koperasi* women, helped local officials to create thematic *kampung* by commercializing their "authentic" Chinese *Benteng* culture to support the mayor's vision for poverty alleviation. Following the instruction to link local economic activities with community-based tourism, *Kampung* Wetan was rebranded *Kampung Tehyan* (*Tehyan* is a name of a traditional musical instrument used to play a local style of music). For that, they built a Chinese-style gate, decorated the *kampung* with red lanterns, and painted a mural with various Chinese images, such as dragons and a pair of people with slanted eyes dressed in Chinese costumes who barely resemble Chinese *Benteng* people. To make it even more "authentic", a *Cokek Sipatmo* dance was performed as the official welcoming dance in front of the oldest Confucian temple to greet visitors. Some women from the *koperasi* have also been appointed as member of government-sponsored local guide groups.

### 2.3. The Postcolonial Feminist Approach to Qualitative Research

Considering the complex relationship between the national conceptualization of womanhood in Indonesia and women's colonial experience of structural oppression, the use of a postcolonial feminist approach in this research is considered well-suited. The birth of postcolonial feminist thinking stems from strong criticism of the hegemonic view of Western feminists, who imposed a narrow perspective on power relations that ignored the dimensions of race, ethnicity, socioeconomic class, and religion that complicate the colonial or postcolonial experience. Postcolonial feminists challenge Western feminists' homogenous representation of women from developing countries, noting the distinct experiences of oppression, injustice, and inequality that are reproduced through colonialism and neocolonialism.

A postcolonial feminist approach proposes a localized and contextualized under-standing of the aforementioned elements. For that reason, it enables us to analyze the

experience of Chinese *Benteng* women through attention to their sociopolitical and cultural history, as well as the structural constraints they experience. Notable postcolonial feminist writers, including Mohanty [31], Spivak [32], and Abu-Lughod [33,34], have produced a number of critical works that address blind spots Western feminist research has failed to address [31–33]. Elaborating on their critiques of their developed countries counterparts, postcolonial feminists underline the interlocking dimensions of race, class, gender, sexuality, and religion that perpetuate oppression and domination. Mohanty [31], for instance, problematizes the singular and composite representations of the "average women from developing countries" that epitomize discussions of Third World countries.

Postcolonial feminist thinking underscores the specific ways in which colonial and postcolonial experience have shaped the history of developing countries. The interplay of various factors, such as class, geographical location, modern state policies, and local-global relationships, is inevitable. These intersections further construct women's identity in many different ways in any historical moment, which often undermines a traditional concept of womanhood [35]. The process of "othering" women from the Global South and East has led to the idea that oppression is something all women from developing countries experience in the same way, regardless of their caste, race, religion, ethnicity, class, and colonial history, in contrast to the nuanced reality of liberal Western women.

Colonial and postcolonial history, as experienced by Chinese *Benteng* women in Indonesia, were shaped by longstanding sociopolitical exclusion, as well as the structural discrimination faced by a large majority of Chinese Indonesians, although the degree of such experience varied according to social class, economic power, ties with Indonesian natives, religion, and location. Throughout the Indonesian history of nation-building, specifically during the pre-independence war and post-independence state development era, apart from their contribution to the economy, Chinese Indonesians were ignored and silenced by the oppressive authoritarian regime, although they later gained increasing attention during the 1998 political reform. Continuing the process of excluding Chinese Indonesians, who were represented as a monolithic ethnic group, Chinese Indonesian women are often depicted as apolitical and ahistorical beings. By challenging simple binary categories through deconstructing boundaries, postcolonial feminist thought is able to address the critical stumbling blocks of the ahistorical universalist framework by focusing its analysis on "the Other" as a subject of research and on the process of othering.

## 3. Materials and Methods

The data obtained in this research came from ethnographic work that was conducted in 2018 in a semi-urbanized settlement called *Kampung* Wetan (a pseudonym) in Tangerang City, Indonesia. As this research aimed to understand gender relations among the Chinese *Benteng* people and their exclusion in the context of a women's empowerment project, the authors choose women as the primary subjects. In terms of their involvement with the *Koperasi* Pelita project, the women are technically segregated into two distinct groups, namely the ones who call themselves *pengurus* (the managing committee) and those who are regular members. Based on ethnographic methodology, the interactions with these women were mostly in the form of participant observation and semistructured interviews, which enabled the authors to understand their perspective and engage in their daily activities [36–39]. The authors also conducted group interviews and gathered some life-story narratives to get an overall picture of the transformations experienced by some of these women. To build rapport with the informants, the authors often offered to help with various tasks such as cooking, household chores, or walking with them to drop off their children at school or to go to a health center. Gradually, as relationships deepened, the authors were invited to attend family gatherings such as weddings, birthdays, and other celebrations.

All of the Chinese *Benteng* women engaged in the cooperative project were married, and the majority married at an early age. Generally, they describe themselves as housewives, although some of them had separated from their husbands and live only with their children. Most of these women relied on small-scale trading activities, such as selling

snacks and ready-made dishes, to finance their daily expenses. Despite there being a number of them who went to junior high school (even if they never finished), the majority had a low education level. Generally, these women lived with their nuclear family or, in the case of separated women, returned to live with relatives. However, there were also cases where women joined their husband's family or lived with a few other families in one house.

Besides the Chinese *Benteng* women as the core research subject, the authors also interviewed a wide range of actors involved in these women's lives. These included not only those who crossed paths with these women as part of a project basis, such as NGO personnel and government officials, but also local residents who were not part of the cooperative project. In the project context, the authors also attended their weekly savings and lending sessions in the government office, participated in various meetings with government agencies, and attended workshops and training. A few times, the authors engaged in community activities such as monthly registration at a community-based health center, vegetable cultivation in a communal urban gardening project, and a youth gathering during the local election. Through such participation, the authors engaged and conversed with roughly 60 people with whom these women engaged at a project level or in their personal lives.

Generally, the interviews lasted 30 min to 1.5 h, and, prior to the interview, the authors had to familiarize themselves with the local residents. Despite the local residents generally being friendly and chatty, they were sometimes intimidated by the presence of a voice recorder. To overcome such hurdles, the authors often had to rely on memorization of key information and perform triangulation with different respondents on different occasions [40,41]. In addition, the authors kept a daily observation log that allowed for reflective synthesis during the research process. Moreover, the ethnographic study was also enriched by a documentary analysis of aspects of the Chinese *Benteng* people's history, sociocultural status, economy, and political structure, which contributed to our overall knowledge of this community. Project reports, proposals, documentary films, and booklets produced by the NGO were also important to analyze, in addition to books, journals, and news articles that were helpful for understanding the public representation of this community.

## 4. Results and Discussion

### 4.1. The Gaze of "State Ibuism": Patriarchal Construction of Indonesia's "Women's Empowerment"

Despite the vocabulary of "women's empowerment" being introduced in the 1990s and promoted by some Indonesian activists and feminist scholars [6], the term itself is not explicitly used in official documents. The discourse of "women's empowerment" slowly entered into Indonesia's development landscape under the "gender mainstreaming" strategy [4,5] in order to increase women's involvement in national development [42]. Despite its radical roots, "women's empowerment" in Indonesia is generally perceived simply as activities to improve women's ability to actively contribute to development. Indonesian women's needs are commonly defined by the state, which decides how women should be involved in development based on national priorities. During the authoritarian regime, some feminist scholars have noted that state-engineered programs were intended to take down the critical and radical women's movement of the previous period [14,42,43], including banning and dissolving liberal organizations such as *Gerwani* (*Gerakan Wanita Indonesia*), the Indonesia's Women's Movement, established in 1950s. This movement affiliated itself with the Indonesian Communist Party and was strongly supported by Indonesia's first president, Kusno Sosrodihardjo, and, according to Wieringa [43] was probably the most advanced and progressive feminist movement in Indonesia at that time. *Gerwani* supported women workers and peasants and strongly criticized government economic policies that caused skyrocketing food prices. By 1957, *Gerwani* had over 650,000 members and had launched various grassroots programs including those seeking to secure educa-

tion for women and land reform, an antipolygamy campaign, and a movement against discriminative marriage laws [44].

The absence of leftist women's organizations in the New Order era was made up for by state-sponsored women's organizations. *Dharma Wanita* is one such organization, comprised of civil servants' wives, that constructed women's roles according to a gendered division of labor that revolves around *kodrat wanita* (Wieringa [45] explains that women's *kodrat* is a religiously inspired code of conduct based on women's intrinsic "nature"), which determines women's role according to their biological capacity [46] and willingness to care [45]. Wieringa [43] stated that such a homogenization of women's identity was incorporated into the *Ibu* identity, instrumentalized by the New Order regime as an official construction of Indonesian womanhood. Such a construction was elaborated as *Panca Darma Wanita* (the five basic obligations of women), which defines women's roles: as a wife, as a procreator of the nation, as a mother and an educator for her children, as a homemaker, and as an Indonesian citizen [14]. This paradigm therefore determines women's capacities, roles, and obligations in relation to the husband, family, community, and society she tirelessly serves, and the state to which she pledges patriotic loyalty.

Besides *Dharma Wanita*, PKK (this has undergone various name changes: in 1962, it stood for *Pendidikan Kesejahteraan Keluarga* (Family Welfare Education); in 1972, it became *Pembinaan Kesejahteraan Keluarga* (Family Welfare Guidance); from 2000, it has stood for *Pemberdayaan dan Kesejahteraan Keluarga* (Empowerment and Family Welfare) is another official vehicle to educate rural and urban women via a local voluntary platform that employs a militaristic state structure to engage them in national development. According to Wieringa (1992), PKK defines ideal womanhood as sacrificing oneself for the interests of a male-dominated household, which is instrumentalized by state-driven policies [43]. This construction creates the notion of dutiful wives and selfless mothers as dedicated supporters of a patriarchal state that focuses on the family as the smallest unit of society—which, as Suryakusuma [14] explains, is an economic unit, a biosocial unit, and an ideological unit that eventually aims to support the state's development objectives. Therefore, the hierarchy of administered by the PKK directly reproduces the state bureaucratic structure, headed by a First Lady at the national level, followed by the wives of regional heads at the local level and the wives of their husbands' subordinates.

Due to the militaristic line of command, PKK's leadership depends on the husbands' position within the state's structure. Wieringa [43] pointed out that it is impossible for active, capable, and dedicated cadres to be in a leadership position or to propose a reformist agenda because the direction is dictated by wives of high-ranking bureaucrats and rarely opened up for discussion. Active and socially dedicated women are officially recruited as volunteers to execute government programs for women and to socialize state-led messages, such as nutrition, family planning, and childcare. In this context, women are considered" unpaid government servants". Popular activities such as healthy baby competitions and immunization provision are executed by the volunteers under the command of local bureaucrats' wives. Additionally, practical skills such as snack making, sewing, and crafting are taught for two purposes, namely as an extension of the role of homemaker and as additional income for the family (the main family income comes from the husband's earnings). Due to its strategic position, PKK is often used as a political vehicle for female candidates to mobilize female voters [47] and has also functioned as a political broker during the election, while ensuring that women remain loyal supporters [48]. Therefore, despite women seeming to be "empowered" through PKK, it does not provide any space for critical discussion of the existing structure that subordinates and disadvantages women.

The operational logic behind PKK and *Dharma Wanita* as state-orchestrated women's organizations is conceptualized as State Ibuism. Suryakusuma stated that this official gender ideology "led to the process of 'domestication', an inclusive concept in that it incorporates not only the economic sphere, but also political, ideological and cultural ones" [49] p. 8. Through State Ibuism ideology, the notion of ideal womanhood makes clear the limits of women's identity as wife, mother, and citizen, which confines them to the domestic

sphere and keeps them separate from men. State Ibuism ideology is responsible for constructing a homogenous identity for Indonesian women that centers around a gendered definition of an "*Ibu*". According to Djajadiningrat [50], *Ibu* means mother and also refers to all women who are married or are of childbearing age, which implies that all women will become mothers. An *Ibu* is expected to maintain her status and uphold the morality of the family. As *Ibu*, women play crucial roles in terms of managing the household and being responsible for the physical and emotional needs of their family members, which includes nurturing and socializing their children so that they exhibit a certain degree of respect and politeness. Women who perform duties within their family also carry out tasks for the state in a controlled form and a depoliticized environment. Therefore, Ibu has become a static and singular identity that homogenizes the complex reality of Indonesian women across ethnic, religious, and socioeconomic categories. This construction is extremely problematic because it subordinates and marginalizes women, while also depoliticizing the women's movement. Blackburn [51] added that Indonesian women are defined in particular ways as citizens with gendered responsibilities according to the state's developmentalist visions through their participation in state-orchestrated organizations.

By refining earlier concepts of *housewifization* (Mies [52]) and *ibuism* (Djajadiningrat [50], Suryakusuma [49]) connected the domestication of women with the official state construction of the ideal Indonesian woman. She argued that, in this construction, women are valued as instruments of the developmentalist state by fulfilling their domestic role according to the ascribed *kodrat*. For that reason, State *Ibuism* operates through the propagation of nuclear family norms that are built in for women and upheld by state-sponsored organizations in which the husband's role is the most important. Through State *Ibuism*, women's consciousness of their subordination in a patriarchal system is limited, which contributes to their docility and willingness to subscribe to a conservative gender role. Suryakusuma concluded that state-interpreted womanhood is not about women's advancement, but instead about maintaining the *ketertiban* (order and control), *pembinaan* ("guidance", implying indoctrination, construction, and management), and *stabilitas* (stability) of the state. Despite significant changes in Indonesia's political landscape after Suharto's resignation, the construction of ideal womanhood through State *Ibuism* prevails. Furthermore, under the "gender mainstreaming" strategy, women's situation is worse because they enter a precarious labor market while still having unpaid domestic responsibilities, as demanded by society.

In the following section, we elaborate on how the State *Ibuisim* ideology operates within the Chinese *Benteng* community through the formation of *Koperasi* Pelita as a platform to "empower" women in *Kampung* Wetan. As a way to govern Chinese *Benteng* women, the creation of *Koperasi* Pelita has depoliticized the radical roots of "women's empowerment" by exoticizing poverty. Furthermore, by making their presence visible through community volunteering activities, the existence of Koperasi *Pelita* has gradually eroded the need to resist, as in 2010, and instead created a new image of a harmonious society that has been projected by a government-sponsored tourism project.

*4.2. The Twin Purpose of "Women's Empowerment" in Koperasi Pelita*

Under the "women's empowerment" rhetoric, the WRC not only supervises the *koperasi*'s financial transactions, but also initiates activities meant to stimulate women's enthusiasm. While the *pengurus* (management team) *koperasi* learn simple bookkeeping, cashbook registration, and financial reporting, the leader is tasked with recruiting new members to generate more capital to loan out. The members frequently attend *audiensi* meetings planned and arranged by WRC. Generally, the *koperasi* leader chooses which members will be present for such activities; some other members are intentionally uninvited because they were seen as "burdensome" or "too busy to engage", including women with toddlers and those with paid jobs. Consequently, it is always the same groups of active women who usually participate voluntarily in the *koperasi*, and they tend to be more

exclusive in comparison to other members. In turn, their active participation has allowed them to increase their importance, social position, and popularity in the community.

The *Koperasi* Pelita has gained significant attention, especially from government officials. Encouraged by the WRC staff, the *koperasi* was able to secure a temporary office in government premises. However, because the *koperasi* is not a government entity, its affiliation with the authorities was seen as a way to borrow authorized legitimacy to attract prospective members. Often, the *koperasi* is considered as the community's representative, and therefore as a sign of women's participation in local development. However, this borrowed legitimacy is not cost-free; the *koperasi* tends to be permissive of interests that regularly mobilize them as volunteers to support the government's agenda. Furthermore, the existence of *Koperasi* Pelita has increased the popularity of *Kampung* Wetan, especially as they claim to have revived the "authentic" *Cokek Sipatmo* dance through their frequent public performances. Due to this, the local government continues to support them, as they create a positive image for the *kampung*, which was previously perceived negatively due to gambling, alcoholism, covert prostitution, and drug trafficking, especially because of a 2010 incident that created a bad impression.

Through the "empowerment" discourse, the *koperasi*'s formation reflects what Cruikshank [53] defined as "technologies of citizenship", which determine how Chinese *Benteng* women are taught to see and understand themselves through a social lens. In *Koperasi* Pelita, they are seen as "worthwhile", "productive", "social", and "active" members contributing to their society, which encourages them to subscribe to a particular gender role through activities that extend women's domestic responsibility to the public arena, e.g., by joining the PKK group and other activities that count as "feminine" work. This model principally defines women as weak economic units who should be empowered by increasing their contribution to the household and eventually to the developmentalist state. In a Foucauldian framework, the argument about women's contribution to development through the *Koperasi* Pelita can be understood from the analysis of power that interprets women's bodies as productive and docile [54], producing gain for their family and society and, most importantly, serving the patriarchal state. Therefore, by deradicalizing "women's empowerment", the WRC instrumentalizes this discourse to guarantee women's contribution to development, which has diverted away from what Kabeer envisioned in 2005 [55]. In addition, depoliticizing "women's empowerment" as a neoliberal vehicle devalues Chinese *Benteng* women according to certain norms of "building responsible communities, prepared to invest in themselves" [56].

### 4.2.1. Becoming "Ibu-Ibu Koperasi" (Women of the Cooperative): The Wifehood–Motherhood Combination

The formation of *Koperasi* Pelita under the narrative of empowering women is best explained through Foucauldian governmental operations. Foucault [57] emphasized that "technologies of the self" connect control and authority, which represent traditional forms of power, on the one hand; and conscience, identity, and self-knowledge as subjectivity, on the other hand [53]. The presence of WRC staff who regularly helped Chinese *Benteng* women contributed to what Miller and Rose [58] defined as the "technique of the self", involving self-regulating the capacities of "normalized" subjects through the power of expertise. Here, "women's empowerment" operates as a particular mechanism through which the WRC "sought to shape, normalize and instrumentalize the conduct, through decision and aspirations of others to achieve the objective they consider desirable". Similarly, Hache [59] associates empowerment with a kind of self-responsibility for personal wellbeing that would bring about self-transformation towards the maximization of personal endeavors. For that, in a "women's empowerment" intervention, power is directed towards the maximization of their self-consciousness and voluntary participation in projects that are of benefit to the whole population.

Compared to other grassroots initiatives that adopt a more radical approach to women's empowerment, the WRC's approach is rather conventional, tending to use a modern framework that undermines the transformative elements of empowerment. At-

tractive language of "participation", which is framed in market terms, has further promoted policies that specifically target women to engage in productive work and actively contribute to development objectives. Therefore, to increase women's contribution to development, "women's empowerment" strategies are promoted to improve effectiveness and enhance development delivery, especially in the context of poverty, polarization, and exclusion [60,61]. Subsequently, the creation of *Koperasi* Pelita instrumentalizes the dominant view of "women's empowerment", which focuses on the individual economic aspect and aims to integrate women into existing "undisturbed" structures [9,62] and realities shaped by the dominant socioeconomic system, without challenging the dominant structure and gender relations. Therefore, many activities of the *Koperasi* tend to reproduce conservative gender roles by focusing on the qualities of being an *Ibu*, or responsible mother and dutiful wife, according to the official gender construct, through popular techniques such as workshops.

Ratna (age 38) was one of the participants of a workshop organized by the *koperasi*. The workshop sessions were called "Who Am I?" and "Family Finances". In this workshop, some basic skills were taught, especially how to manage a household efficiently and effectively, including creative ways to obtain extra income. The topic of how to be a "good financial manager" for the family was a popular one, as many women often complain because their husband's income is insufficient. In the workshop, the trainer repeated that women have to be smart about managing their husbands' income because it is their responsibility. At the same time, the workshop normalized men's responsibility as the main breadwinner. By relying on specific gendered qualities, such as "discipline in saving" and being "smart about spending", participants were taught how to productively use such qualities to be better financial managers for their family. Additionally, participants learned about practical skills and were informed about extra support that they could access through government social welfare programs for low-income families. Ratna summarized these qualities as meaning that one "should be smart about administering income" and "must know many tricks" to strategize about the daily financial situation. These are crucial qualities required to be an efficient "financial manager". According to their *kodrat* as wives and mothers, women must prioritize their family before themselves, and what they have earned is secondary to their husband's earnings. Moreover, as responsible mothers, they were continuously reminded about their role in ensuring their children's upbringing by emphasizing the care functions that they should selflessly perform. This means sacrificing their desires and aspirations and indicates that selflessness is an important quality and that the family should be the first priority.

Through the *Koperasi* Pelita, these women were continuously reminded by their facilitators to prioritize their family responsibilities despite being active in the *koperasi*. Such *subjectification* conforms to the social construction of their *kodrat* as selfless mother and compliant wife by underlining that domestic unpaid labor should be prioritized over other activities. Therefore, the existence of the women-run *Koperasi* Pelita fits in with a common narrative of apolitical "women's empowerment" and is part of the collective idealization of Indonesian womanhood that stems from a particular image of an *Ibu*. This collective image of the "empowered" Chinese *Benteng* woman rests on her ability to master both worlds: domestic unpaid work and community voluntary activities that involve femininized tasks such as better family planning, improved nutrition, children's education, basic health services, and cultural preservation. As a result of this prioritization, women are taught to abandon their own aspirations; this ultimately perpetuates conservative gender roles.

Due to the absence of individual support, some women had to cease their participation in *Koperasi* Pelita despite their initial enthusiasm for the activities. Some of these women decided to stop mostly because of an increased load of unpaid domestic work, especially for new mothers, which made them less able to attend gatherings and meant they were gradually left behind. Their strategic position as representatives in the *koperasi* was then lost as they were replaced by other women who were "fully committed" without "interruption". Mariah (age 30), for instance, wanted to engage in the *koperasi*'s activities but needed help

taking care of her children. As much as she wanted to continue, she could not rely on her mother's assistance because she was already occupied with her sister's children and a porridge stall. Mariah once hoped that the *koperasi* would teach her more practical skills, such as operating a computer, which would be useful for applying for jobs. Nevertheless, in the process of empowering women, males are left out of the equation because their role as the main breadwinner means they are considered too busy to engage in the project.

The practice of limiting a women's role outside the house emerged as the politics of *housewifization* became entrenched more deeply in Indonesian society. Indonesian women were primarily seen as wives and mothers based on their biological capacity. The creation of the ideals of the responsible mother and dutiful wife in the "women's empowerment" model operated by *Koperasi* Pelita reflects what Foucault described as an intention to create and to maintain "rules of law, the technique of management, and also the ethics, the ethos, the practice of self, which would allow these games of power to be played with a minimum domination". For more than three decades, during the authoritarian regime, Indonesian women were seen as subjects and compliant ancillaries of their husband by state-sponsored projects such as *Dharma Wanita* and PKK. However, more than two decades since the regime ended, such a construction remains. Women are no longer conceived as docile subjects, but they are still seen as rational subjects who voluntarily perform their *kodrat* as mother and wife.

### 4.2.2. The New Image of the "Ibu-ibu Koperasi": Active and Contributing Member of Society

The current sociocultural construction still focuses on women's *kodrat* and underlines women's contribution to Indonesia's development trajectory. As good citizens, women are constituted as autonomous agents of an advanced, liberal society and can productively contribute to a system that can generate particular benefits for the state and society. As *Koperasi* Pelita is formally located in a government office, its participants are familiar with how the officials work. An active woman from the *koperasi* might be treated by government officials as one of their own. Women are also permitted to use government office equipment and stationery for *koperasi* administration purposes. Since the *koperasi*'s presence and activities have contributed to a positive image for the whole neighborhood, these women are treated favorably by government officials. They start to build more harmonious engagements, which, interestingly, are more of a mutually beneficial patron-client type of relationship than a community-led organization that works for the community's interest. These active women are treated as a "shadow apparatus", in that they enjoy a certain privileged status as they frequently represent the women of *Kelurahan* Kembangan on various occasions.

With help from WRC staff arranging *audiensi* with the district's sectoral offices, these women are accustomed to dealing with government officials. This is like climbing up a social ladder; these active women are perceived by other women in their neighborhood as becoming more important through their intimate knowledge of the government. Their sense of pride is often associated with how frequently they wear a batik blouse to attend a formal meeting with the government. According to Linawati (age 42), WRC staff often warn them to wear something appropriate for *audiensi* and told them to stop wearing short pants, flip-flops, or T-shirts because these are inappropriate. Linawati said that once a member of the NGO staff snapped at her because of her inappropriate outfit and warned her that this was not a trip to the market for which she could dress sloppily. These women then ordered a batik uniform to wear whenever they had an important meeting to attend, especially with the government. *Audiensi* with government agencies are not regular but occur only when the WRC receives project funds that can be spent on small allowances and lunches for these women. Unfortunately, not all participating women were aware of the objective of the *audiensi*; the majority of them were happy enough to tag along without knowing the reason behind it and did not bother to demand further information. Some of them were happy to have the chance to sit and listen, to get to know more people, to obtain new information, or just to get out of the house.

"Women's empowerment" rhetoric has become ubiquitous, but as a development buzzword, its meaning has been diluted, as it is widely used to stand in for nonbinding goals that diverse groups can be seen to support without actually subscribing to any specific feminist principles [63,64]. Feminist scholars have long argued that the empowerment framework has indeed instrumentalized feminist language to legitimize neoliberal policy goals pursued solely via economic participation. Calkin [65] coined the term "empowerability" to refer to the specific skills acquired through a neoliberal mode of development that requires a series of "activations". Through the *Koperasi* Pelita, the WRC performed this "activation" process on Chinese *Benteng* women through different initiatives that construct particular roles, such as "dutiful wife", "responsible mother", and "contributing member of society", that fit the State *Ibuism* ideology's ideal of womanhood. Following this ideology, the narrative of women's empowerment employed here is based on particular neoliberal women to whom the term "empowerability" would be applied. Therefore, in this context it can be argued that empowered women are socially constructed based on the dual notions of femininity and productivity according to state-constructed State *Ibuism*, which maintains that their domestication is a necessary contribution to the national development agenda.

Neoliberal development projects that specifically target economically disadvantaged women use the depoliticized language of "women's empowerment" to maximize women's economic productivity while preserving *Ibu* as the backbone of family morality. Under *Koperasi* Pelita, empowerment activities that target women's practical needs are simply a way to symbolically include women in national development schemes that reproduce socially constructed gendered roles without actually challenging the gender status quo. For NGOs such as the WRC that implement community development by capitalizing on the "women's empowerment" approach, common activities to involve these women center on creating understanding and capacities according to the prevailing gender ideology imposed upon them. As a result, women may consider practical needs such as healthcare facilities for children and pregnant mothers or practical skills for generating additional income as "pro-women" in that they seem to indicate "gender inclusivity" in development planning; in fact, such plans are opposed to women's strategic interests, namely, a critical education that would allow them to challenge deep-seated gender inequality. Just as Sholkamy [66] argued, alleviating poverty and enabling women to earn an income can improve their lives, but an enabling environment that affirms the right to work, to property, to safety, to a voice, to sexuality, and to freedom is not created by sewing machines or microcredit alone. Therefore, projects that aim to alleviate women's poverty through empowerment require a paradigm shift that enables women to challenge the social regime that is subordinating them, as well as gender stereotypes and the gendered division of labor.

## 5. Conclusions

Suryakusuma's State *Ibuism*, the backbone of Indonesia's gender ideology, institutionalizes women's duties to altruistically serve their husbands, families, communities, and the state. The role that women play in the longevity of society is maintained to lay the foundation of the "state family" [67] concept, which underlines women's core domestic function by linking both motherhood and wifehood to gendered citizenship, following the patriarchal order of the state [68,69]. Emphasizing the desired qualities of womanhood in Indonesian society affirms women's responsibility to uphold family morality, which is expected to lead to greatness for the nation. Brenner [70] pointed out that women's *kodrat* is key to maintaining such vigilance.

The ascendency of neoliberal development has led to many vulnerable households being trapped in a vicious cycle of poverty. The widening gap due to economic hardship because of market-based public policies has meant that women must play a dual role. Feminist scholars such as Abu-Lughod [33] have contended that the neoliberal approach to development centralizes the problems of poverty because of a lack of engagement with the market as the only domain where individual freedom and rational responsibility are seen as solutions. Therefore, a shift towards deradicalized and depoliticized women's

empowerment serves the purpose of maintaining women's *kodrat* within State *Ibuism* as a formal gender ideology and a way of curbing resistance to subordination that is perpetuated by neoliberal development policies. Reflecting on this, we can see that a short-term intervention, such as the microfinance initiative promoted by the WRC, is an appealing activity for Chinese *Benteng* women in *Kampung* Wetan and seems like a way to "empower" them. Through the so-called "inclusive cooperative", these women are expected to play a crucial role as a secondary breadwinner by engaging in economic activity in a productive and responsible way, as they now have better access to financing. Consequently, this model perpetuates their marginalization and exclusion by forcing them to take responsibility for their poverty, rather than questioning the unequal structures of society as well as the exploitative mechanism of the capitalist market.

The depoliticization and deradicalization of women's empowerment in a neoliberal development context means teaching women to be disciplined and responsible mothers, wives, members of society, and citizens according to their *kodrat*. Indonesia's understanding of "women's empowerment" is filtered through the lens of State *Ibuism*, which thoroughly explains the gendered relationship between the state as the patriarchal social order and women as the subjects to be governed. The idealized figure of the *Ibu*, primarily understood as the "faithful wife" and "dutiful and supportive companion", and secondarily as a "responsibly loving mother", has constrained women to uphold their gendered virtue [71]. To maintain such integrity, the notion of respectable Indonesian womanhood is cultivated through marriage, most importantly a legal marriage, which paves the way to wifehood and motherhood. In this case, women's empowerment is seen as a corrective activity with therapeutic value rather than a radical strategy to overcome the oppression of women. This would require partnership between some not-so-radical women's organizations [72], such as the WRC, which focus less on pursuing reformist ideas of gender equality and more on cultural patterns. This is what Wieringa [73] has argued: the introduction of the idea of gender harmony emphasizes mutual understanding between men and women without creating a conflict. In that context, the superficial women's empowerment activities implemented by the WRC, which emphasize material self-reliance measured in economic terms, are less likely to disrupt the existing gender status quo. *Koperasi* Pelita has become a domain of governance whereby a market-driven society, introduced by women's NGOs as brokers and backed up by the patriarchal nature of the state, has instilled in Chinese *Benteng* women the need to be reliable, dutiful, and obedient housewives as well as devoted, caring, and responsible mothers. Women are also expected to be socially active members of Indonesian society, exhibiting high levels of volunteerism and commitment to the state apolitical project. The *Koperasi* Pelita, therefore, as a female-managed financial organization, is not only a governmental operation, but also a site of domination where the gender order is recreated based on the ideology of State *Ibuism*, which is maintained and reproduced under the patriarchal ideology of the state.

Having said that, we do not think the economic aspect should be removed entirely from the project of women's empowerment. However, what might be useful is to strategically employ the economic dimension to move beyond income-generating activities that are built based on a gendered division of labor. We call for greater reflection to understand how imbalances in the local power structure have shaped the identities and subjectivities of women and are profoundly embedded in society. By engaging in a discussion of how power relations disadvantage women, we might re-orient women's empowerment into a more politically transformative idea that would challenge not only the patriarchy but also the broader social system, which includes class structure, race, ethnicity, caste, and religion [7,9]. Such a discussion would enable women to critically analyze the main causes of their subordination in the current system, form strategic alliances to challenge patriarchal structures, and advocate for women's interests.

Through forming strategic alliances between women's groups in Indonesia to pursue the long-term vision of gender equality and justice, repoliticization of women's empowerment is more likely to materialize. MAMPU, an eight-year-old program commissioned

by an Australian aid agency in partnership with a grassroots women's organization in Indonesia, is an example of a project that embraced the women's empowerment approach to advocate for women's strategic interests in various arenas. By focusing on five main themes—namely, access to a social safety net, labor conditions, migrant workers, health and nutrition, and gender-based violence—the MAMPU program enabled these grassroots women's organizations to actively engage in key decision-making arenas to influence policies and practices that affect their needs and interests. Unfortunately, an initiative such as MAMPU is extremely dependent on donors' financial interests and availability, which could potentially jeopardize all the good work that has been done during the program. For that reason, internationally driven women's empowerment projects are often challenged on the question of sustainability at the local level once the project financing is complete. Lastly, unless feminist alliances are able to secure ongoing resources, projects such as MAMPU that attempt to repoliticize women's empowerment and focus on women's strategic interests on the basis of critically challenging the existing power structure that disadvantages women in the first place will struggle to effect real societal change.

**Author Contributions:** Conceptualization, V.F.H., J.P.R., and G.L.-M.; methodology, V.F.H., J.P.R., G.L.-M., and V.M.-S.; formal analysis, V.F.H., J.P.R., and G.L.-M.; investigation, V.F.H. and J.P.R.; data curation, V.F.H. and J.P.R.; writing—original draft preparation, V.F.H. and J.P.R.; writing—review and editing, V.F.H., J.P.R., and G.L.-M.; visualization, V.F.H., J.P.R., G.L.-M., and V.M.-S.; supervision, V.F.H., J.P.R., G.L.-M., and V.M.-S.; funding acquisition, V.M.-S. All authors have read and agreed to the published version of the manuscript.

**Funding:** This research received no external funding.

**Institutional Review Board Statement:** Not applicable.

**Informed Consent Statement:** Not applicable.

**Conflicts of Interest:** The authors declare no conflict of interest.

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
