# Peer review of "State Ibuism and Women’s Empowerment in Indonesia: Governmentality and Political Subjectification of Chinese Benteng Women"

_sustainability, doi:10.3390/su13063559_

Round 1
Reviewer 1 Report
This is an interesting, well-argued and well-written paper, a powerful criticism of "empowerment" in Indonesia. It demonstrates an excellent understanding of literature on women and development. In addition, it appears that the authors have an intimate understanding of local conditions in Indonesia. I appreciate the focus on Indonesia's "Other"--Chinese Indonesians, and a powerful critique of neoliberal development models. Several recommendations are as follows:
1) On p. 8, when discussing Ibuism, the authors mention that this official gender ideology leads to "depolitization" (among other things), and identify "citizen" as one of women's roles (linked to depolitization). This is a bit confusing because citizenship is usually linked to political sphere. It would be useful to explain this interpretation of depolitization.
2) The authors assume that their reader will be familiar with different approaches to women, gender and development (such as "Women and Development," or "Gender and Development"). It may be useful to briefly explain them, perhaps in a footnote, and explain their relationship with the neoliberal model criticized in this paper.
3) The authors seem to suggest that an alternative to "empowerment" supported by neoliberalism is opposition to gender subordination. It may be helpful to outline this vision in a bit more detail. How can it be pursued in Indonesia? By whom? What about internationally?
4) The authors suggest that they spent a lot of time doing fieldwork; however, the essay does not include much insights from the fieldwork. It may be useful to include more observations from the field in order to create a more realistic picture of what is going on the ground.
Despite these observations, I think that this is a good paper, an excellent addition to the special issue.
Reviewer 2 Report
The article complies with the structure and scientific rules associated with the creation of science. The article presents an innovative vision and results that contribute to the progress of science.
The methodology used is adapted to the object of study and is well developed and justified.
The main improvement can be made in a reinforcement of the theoretical framework on feminist studies / feminist theories, which seem to me little developed in the theoretical part. What similar studies do you present? Namely with the use of the same methodology (ethnography).
The discussion and presentation of results are clear and represent an improvement in this thematic.
